# The Individual Division of Food Hoarding in Autumn Brandt’s Voles (*Lasiopodomys brandtii*)

**DOI:** 10.3390/ani14182719

**Published:** 2024-09-20

**Authors:** Zhiliang Zhang, Fan Bu, Shanshan Sun, Ming Ming, Tao Liu, Yanan Li, Xiaodong Wu, Xueying Zhang, Shuai Yuan, Heping Fu

**Affiliations:** 1College of Grassland, Resources and Environment, Inner Mongolia Agricultural University, 29 Erdos East Street, Saihan District, Hohhot 010011, China; zhangzhiliang1135@163.com (Z.Z.); bufanimau@163.com (F.B.); sunshanshan557@163.com (S.S.); bai_mingming2023@163.com (M.M.); ltao12358@163.com (T.L.); lyn15648168956@163.com (Y.L.); wuxiaodong_hgb@163.com (X.W.); 2Key Laboratory of Grassland Rodent Ecology and Rodent Pest Control at Universities of Inner Mongolia Autonomous, 29 Erdos East Street, Saihan District, Hohhot 010011, China; 3Key Laboratory of Grassland Resources, Ministry of Education, 29 Erdos East Street, Saihan District, Hohhot 010011, China; 4State Key Laboratory of Integrated Pest Management, Institute of Zoology, Chinese Academy of Sciences, Beijing 100101, China; zhangxy@ioz.ac.cn

**Keywords:** Brandt’s voles (*Lasiopodomys brandtii*), division of labor, exploratory behavior, food-hoarding behavior, spatial memory

## Abstract

**Simple Summary:**

The division of labor and cooperation are typical behavior patterns of gregarious mammals, but it is unclear whether Brandt’s voles exercise a division of labor before overwintering. We observed and recorded the behavioral activities of Brandt’s voles (*Lasiopodomys brandtii*) with an infrared camera and conducted behavioral experiments on individuals with a food-hoarding division. We found that Brandt’s voles had two types of hoarding behavior: high food hoarding and low food hoarding. Furthermore, high-food-hoarding individuals had greater spatial memory. The individual division of labor during the hoarding period of Brandt’s voles was analyzed, and the personality characteristics of individuals were measured. The goal was to better understand the division of labor among different individuals in hoarding, as well as the emergence and significance of the social division of labor in gregarious animals.

**Abstract:**

Brandt’s voles (*Lasiopodomys brandtii*), one of the main non-hibernating rodent species in the typical grassland of Inner Mongolia, live in groups and have the behavioral habit of hoarding food in underground warehouses in autumn to prepare for the winter food shortage ahead. The division of labor and cooperation are typical behavior patterns of gregarious mammals, but it is unclear whether Brandt’s voles exercise a division of labor in food hoarding before overwintering. To explore the division of food hoarding in Brandt’s voles during the autumn period, three treatments, namely added food, added food + competition, and control, were set up with three replicates. An infrared camera was positioned to observe and record the behavior of Brandt’s voles under different treatments. Next, behavioral experiments regarding food-hoarding division were performed on individuals. The results showed that (1) Brandt’s voles had two types of hoarding behavior, namely high food hoarding and low food hoarding, but not all individuals displayed hoarding behavior. (2) In all treatments, feeding behavior, which was the most important type of behavior, accounted for the highest proportion of all behaviors. (3) There was no significant difference in body weight and sex between high- and low-food-hoarding individuals of Brandt’s voles, and there was no significant difference between high- and low-food-hoarding individuals in other divisions of labor either. (4) There was no significant difference in inquiry ability between high- and low-food-hoarding groups, but there was a significant difference in spatial memory. High-food-hoarding individuals had greater spatial memory. In summary, Brandt’s voles had two types of hoarding behavior: high food hoarding and low food hoarding. Furthermore, high-food-hoarding individuals had greater spatial memory.

## 1. Introduction

Many animals have evolved food-hoarding behavior in nature due to the lack of food resources and the unpredictable changes. This enables them to obtain and store food through periods when food resources are abundant so they have enough food to survive during the periods of food resource shortage [1]. For rodents living in the high latitudes of the Northern Hemisphere, the harsh environment in winter is the main challenge for their survival [2]. During winter, animals use stored food to overcome food scarcity and to have enough energy for reproduction, which is of far-reaching significance for the survival of the population [3]. Rodents are the main group of mammals that store food [4]. Animals that practice hoarding behavior can mitigate the adverse effects of food shortage with their hoarded food, which helps them overcome the stress of energy shortage during the cold winter [5]. Conversely, animals that do not practice food-hoarding behavior need to migrate; use body energy reserves (such as body fat); or enter a state of metabolic inhibition, such as hibernation, etc., when they are short on food [6]. Some of these animals do not store food and instead meet their own needs by stealing the food stored by other individuals [7].

Rodents are the most abundant food-storing group in mammals [8], and they include prairie voles (*Microtus ochrogaster*) [9], Brandt’s voles (*Lasiopodomys brandtii*) [10], and Mongolian gerbils (*Meriones unguiculatus*) [11]. Although rodents tend to share similar hoarding behaviors, the amount of food stored by different individuals of the same species may vary greatly [12]. In field studies, a large difference was found in food-hoarding quantity within the North American pika (*Ochotona princeps*) [13] and North American red squirrel (*Tamiasciurus hudsonicus*) [14]. In another example, Mongolian gerbils show a dimorphic food-hoarding behavior depending on whether food is abundant or in short supply; that is, some individuals hoard large amounts of food, while others hoard almost none [5,15]. Sex differences in rodents have also been shown to impact food-hoarding behavior [16]. Generally, animals can improve their survival rate under food shortage conditions when they are in the habit of storing food. However, there is a trade-off between coping with risks and obtaining more food resources [17]. More and more studies have found that animals’ personality characteristics are related to their hoarding strategies [18]. For example, individuals with a history of food hoarding usually have greater inquiry ability in golden hamsters (*Mesocricetus auratus*) [19], while the individuals that hoarded greater amounts of food have greater exploration ability in North American red squirrels [20].

Brandt’s vole (Rodentia, Cricetidae, Arvicolinae) mainly lives in the Inner Mongolia steppe; eastern Mongolia; and the southeast of Transbaikal, Russia. It is a typical non-hibernating gregarious steppe rodent. In winter, a Brandt’s vole mainly feeds on grass in cave warehouses [21]. In the field, Brandt’s vole was observed to begin the hoarding period in mid-August, when it stopped breeding. During the peak period of food hoarding, Brandt’s voles had a higher frequency of activity and food hoarding than during other periods [22]. A large number of studies have been carried out on the feeding and hoarding behaviors of Brandt’s voles, including on their overwintering hoarding of Artemisia plants [23], their feeding and hoarding processes, feeding distance [24], selection preference [17], foraging area [25], and the relationship between vegetation conditions and their hoarding behaviors [7]. However, few reports have considered the individual division of food hoarding within a Brandt’s vole family during the hoarding period. Cooperation and division are typical behavior patterns in gregarious mammal species. Here, we hypothesized that (1) Brandt’s voles have an individual division of behavior during the hoarding period; (2) the frequency of food hoarding may be related to individual weight, sex, and personality; and (3) the addition of food and competition may affect the food hoarding of Brandt’s voles. To test this hypothesis, we studied the hoarding behavior of Brandt’s voles in the typical steppe area of Inner Mongolia. The individual division of labor during the hoarding period of Brandt’s voles was analyzed, and the personality characteristics of individuals were measured to verify the above assumptions. The goal was to better understand the division of labor among different individuals in hoarding, as well as the emergence and significance of the social division of labor in gregarious animals.

## 2. Materials and Methods

### 2.1. Animals and Sample Treatments

The study area is located in the Inner Mongolia Autonomous Region Xilin Gol League East Ujimqin Banner’s “Inner Mongolia Agricultural University typical steppe ecology and rodent control research base” (E 115°30′–116°30′, N 44°42′–45°15′). The site is a typical steppe with a grass layer height of 10–20 cm. The main plant species are Manyroot onion (*Allium polyrhizum*), grand needlegrass (*Stipa grandis*), and *Neopallasia pectinata*. The rodents in this area include the Brandt’s vole, striped hamster (*Cricetelus barabensis*), Daurian ground squirrel (*Spermophilus dauricus*), and Mongolian gerbil, among which Brandt’s vole is the dominant species.

### 2.2. Experimental Design

In the hoarding season, September 2022, three treatments were set up in the typical steppe study area, including added food, added food + competition, and control; each treatment had 3 replicates.

Added food: Peanuts were placed in the central area of the cave system to enable observation of the hoarding of peanuts in Brandt’s voles.

Added food + competition: Peanuts were placed between the two burrow systems, and the hoarding of peanuts by Brandt’s voles was observed in one of the burrow systems.

Control: There were no peanuts, and the hoarding of Brandt’s voles was observed in its natural condition.

Peanuts were selected as the supplementary food, and a quantitative feeding method was used. Briefly, 100 g of peanut was placed in the added food and the added food + competitive hole system treatments in the morning, middle, and evening (300 g per day and continuously for 4 days). Infrared cameras were set up around the cave system to record the hoarding behavior of Brandt’s voles. The treatments were set up 200 m apart from each other to ensure that there would be no interference between them.

### 2.3. Marking Brandt’s Voles

Before the start of the experiment, Brandt’s voles were captured in the experimental cave system with traps. Newly captured individuals were stained with animal hair dyes on different parts of their heads, backs, and buttocks for individual identification. Captures continued for 4–5 days until the captures only resulted in marked individuals. For each capture, number, weight, sex, and other basic information were recorded for the Brandt’s voles captured in the different cave systems.

### 2.4. Monitoring Hoarding Behavior

After marking Brandt’s voles, every burrow system was fitted with an infrared camera, which led to a total of nine cameras. The cameras were installed 30~40 cm above the ground. The camera was tilted slightly downward in the horizontal direction to monitor the behaviors of Brandt’s voles, including food hoarding and vigilance. The parameters of the cameras were adjusted to video mode, with 24 h monitoring, and each video shooting time of 60 s. The cameras were installed at 6:00 in the morning and continued to shoot video for 4 days. To ensure that the camera battery was at a sufficient level, it was checked repeatedly during the shooting period. During the battery replacement period, the video data on the infrared camera were imported into the computer, and the recording was numbered and archived. The video captured with the infrared cameras was watched, and the behavioral activities were recorded and analyzed in each video.

### 2.5. Video Data Analysis

The behavioral spectrum parameters of Brandt’s voles in this study were defined in accordance with the description previously [26]. They are presented in Table 1.

Through watching all the videos captured by the infrared camera, the frequency of food hoarding, vigilance, and feeding behaviors of Brandt’s voles were counted for each day. Each time the voles carried food to the nest was counted as one instance of food hoarding. The frequency of food hoarding is the sum of the frequency of food hoarding in 4 days. By counting the food-hoarding frequency of Brandt’s voles for 4 days, it was found that the food-hoarding frequency of high- and low-food-hoarding individuals was significantly different. The food-hoarding frequency of high-food-hoarding individuals was dozens of frequencies, while that of low-food-hoarding individuals was only a few frequencies. Therefore, the intermediate number of 10 was taken as the division standard. The behavioral characteristics of high- and low-food-hoarding individuals were determined according to the frequency of food hoarding.

### 2.6. Personality Determination

The videos were screened to identify high- and low-food-hoarding individuals, and the cages were used to capture marked Brandt’s voles that had appeared in the video and bring them back to the laboratory for open-field experiments, in which they participated one at a time. The animals were allowed to enter the open field 10 min before the experiments started to allow them to adapt to the new environment. After the adaptation period, Brandt’s voles were recorded on video for 10 min. After the open-field experiment, Brandt’s voles were restricted from food for 24 h, and then the Y maze experiment was performed. At the end of the second experiment, all animals were returned to their original home system.

### 2.7. Open-Field Test

The cages were used to live-capture the marked Brandt’s voles that had appeared in the video, and they were taken back to the laboratory for open field and Y maze tests. The open-field test is a general method for determining the exploratory ability of rodents [27]. The experimental device is an opaque acrylic plate box (60 × 60 × 50 cm) that is marked with a grid of 4 × 4 cm squares at the bottom to distinguish between the central area and the peripheral area(Shanghai Xinru, China). The middle 4 squares are considered the central area, and the remaining 12 squares are considered the peripheral area. Ten minutes before the experiment, the animal undergoing testing entered the open field so it could adapt to the new setting, and then for the next ten minutes, as the experiment was run, the Brandt’s vole was video-recorded. The experiment was repeated for each individual. The amount of time that the animals spent in the center squares and the number of times they crossed the center were recorded. The longer an animal stays in the center, and the more times it crosses the center, the greater their inquiry ability, and vice versa [28].

After each iteration of the experiment was completed, the device was wiped with 75% alcohol to avoid any influence of the previous individual’s feces, urine, or odor on the next individual’s test results.

### 2.8. Y Maze Test

The Y maze measures the spontaneous exploration behavior of animals and is used to evaluate learning and memory function [29]. It is the best and most convenient method for detecting working memory. The device consists of three identical arms (40 × 8 × 30 cm): the novel arm, the starting arm, and the food arm. Each arm is set at an angle of 120° from the others, and there is a detachable partition in the center(Shanghai Xinru, China). Different shapes of geometric marks are attached to each arm for visual identification. The animals fasted for 24 h before the experiment, and then they entered the training phase. During training, only the starting arm and the food arm were opened, and peanuts were put in the food arm. After 5 min of training, the animals and peanuts were taken out, and the residual odor was removed with 75% alcohol. After 1 h, the test phase began. This time, no food was present, and the three arms were opened, the Brandt’s voles were placed at the starting point of the start arm and allowed to move freely. For 10 min, the Brandt’s voles were video-recorded. From these recordings, the time and number of shuttles of each animal in each arm were determined; the greater the percentage of time spent in the novel arm or the greater the percentage of the number of times entering and exiting the novel arm, the better the memory. The opposite would indicate poor or impaired memory [30].

After each iteration of the experiment was completed, the device was wiped with 75% alcohol to avoid any influence of the previous animal’s feces, urine, or odor on the next animal’s test results. At the end of the behavioral tests, all animals were returned to their original home system.

### 2.9. Statistical Analysis

SPSS 26.0 (SPSS Inc., Chicago, IL, USA) was used for statistical analysis. Before statistical analysis, all data underwent Shapiro–Wilk and Levene testing. The correlation between food-hoarding frequency and body weight was analyzed. The nonparametric Kruskal–Wallis rank-sum test was used to compare the differences in food hoarding among the caves in the treatment. The independent-sample *t*-test was performed on the behavior frequency and sex of the high- and low-food-hoarding groups, with the group as the fixed factor. The Mann–Whitney *U* nonparametric test was used to analyze the data of behavioral indicators. All results are expressed as mean ± SE, at a significance level of *p* < 0.05. The entire visualization was performed with Origin 2021.

## 3. Results

In September 2022, 90 Brandt’s voles (9 burrows) were captured and marked. The infrared camera video-recorded 56 individuals displaying hoarding behavior (Table 2).

### 3.1. The Frequency of Each Behavior

Throughout the study, 4717 effective videos were captured using infrared cameras, and the main behavior types displayed by Brandt’s voles in the recordings were food hoarding, sentinel, avoiding, and fighting. The hoarding behavior of Brandt’s voles was significantly higher than all other behaviors in all treatments (added food: *Z* = −2.603, *p* = 0.009; added food + competition: *Z* = −2.617, *p* = 0.009; control: *Z* = −2.636, *p* = 0.008), and there was no significant difference between other behaviors. The food-hoarding behavior was significantly higher in the added food and added food + competition treatments than in the control treatment (added food: *Z* = −2.699, *p* = 0.007; added food + competition: *Z* = −2.234, *p* = 0.025) (Figure 1).

### 3.2. The Difference in Behavior Frequency in High- and Low-Frequency Hoarders

When comparing the differences in each behavior between the different food-hoarding groups, there was no significant difference between high- and low-food-hoarding groups in sentinel, avoiding, fighting, and other behaviors (sentinel: *t* = 0.496, *df* = 14, *p* = 0.628; avoiding: *t* = 0.905, *df* = 9, *p* = 0.389; fighting: *t* = 0.775, *df* = 3, *p* = 0.495; others: *t* = 1.643, *df* = 12, *p* = 0.126) (Figure 2a). There was also no significant difference between the high- and low-food-hoarding groups in the added food + competition treatment (sentinel: *t* = −1.512, *df* = 11, *p* = 0.159; avoiding: *t* = −1.000, *df* = 6, *p* = 0.356; fighting: *t* = 2.535, *df* = 5, *p* = 0.052; others: *t* = −0.531, *df* = 8, *p* = 0.610) (Figure 2b) or in the control treatment (sentinel: *t* = −0.655, *df* = 6, *p* = 0.537; avoiding: *t* = 0.775, *df* =3, *p* = 0.495; fighting: *t* = −1.000, *df* = 2, *p* = 0.423; others: *t* = −0.158, *df* = 7, *p* = 0.879) (Figure 2c).

### 3.3. Differences in Food Hoarding by Brandt’s Voles in Various Treatments

The analysis of the hoarding frequency of Brandt’s voles in various treatments shows that not all Brandt’s voles were involved in hoarding, and there was no significant difference in the hoarding frequency of Brandt’s voles among the three burrows in each treatment (*p* > 0.05) (Table 3). By comparing the relationship between the frequency of food hoarding and body weight in different treatments, it was found that the correlation between the frequency of food hoarding and body weight of Brandt’s voles was not significant (*r* = 0.041, *p =* 0.765) (Figure 3).

### 3.4. Sex Differences between High- and Low-Frequency Food Hoarders

There was no significant difference in the frequency of food hoarding between males and females in the high-food-hoarding group in the added food treatment (*t* = −0.445, *df* = 11, *p* = 0.665), the added food + competition treatment (*t* = 0.311, *df* = 7, *p* = 0.765), or the control treatment (*t* = 0.469, *df* = 4, *p* = 0.664). There was also no significant difference in the frequency of food hoarding between males and females in the low-food-hoarding group in the added food treatment (*t* = −0.889, *df* = 11, *p* = 0.393), the added food + competition treatment (*t* = 0.368, *df* = 8, *p* = 0.723), or the control treatment (*t* = −0.600, *df* = 3, *p* = 0.591) (Figure 4a–c). The three treatments did not have any significant differences in male and female individuals between the high- and low-food-hoarding groups, according to a further comparison (Figure 4d–f).

### 3.5. The Exploratory Behavior in High- and Low-Frequency Food Hoarders

The results of the open-field experiment showed that there was no significant difference in the frequency of crossing the center (*Z* = −0.708, *p* = 0.479) or the time spent in the center (*Z* = −1.292, *p* = 0.196) between high- and low-frequency food hoarders (Figure 5a,b).

### 3.6. Spatial Memory in High- and Low-Frequency Food Hoarders

The results of the Y maze test showed that there was no significant difference in the time spent in the starting arm (*Z* = −0.467, *p* = 0.640) and the time spent in the food arm (*Z* = −1.173, *p* = 0.241) in the Y maze between the high- and low-frequency food hoarders (Figure 6b,d). However, the frequency of crossing the starting arm (*Z* = −2.274, *p* = 0.023), the frequency of crossing the food arm (*Z* = −1.986, *p* = 0.047), the frequency of crossing the novel arm (*Z* = −2.154, *p* = 0.031), and the time spent in the novel arm (*Z* = −2.240, *p* = 0.025) (Figure 6a,c,e,f) were all significantly higher in the high-food-hoarding individuals than in low-food-hoarding individuals.

## 4. Discussion

Animals usually store food when food resources are abundant. This behavior ensures that animals will still have enough food when food resources are lacking or under greater competitive conditions. Hoarding especially alleviates the difficulties caused by food shortage seasons. Food not only affects the development, reproduction, and behavior of rodents, but it also affects the survival of their offspring [31,32], reduces the risk of predation, and determines the density of rodent populations [33]. Mammals that live in communities often cooperate and divide labor in cluster activities. Cooperation is a common feature in gregarious animals such as Mongolian gerbils [34] and meerkats (*Suricata suricatta*) [35]. Animals cooperate to ensure the development of groups via childrearing, foraging, and whistle-blowing [36]. The division of different individuals in gregarious rodents may play an important role in cooperative hoarding. A study on an ant colony through the long-term automatic tracking of behavior and mathematical modeling demonstrated that when the size of a social group reaches six individuals, there will be a division of labor within the group [37]. The indoor breeding population of Mongolian gerbils displayed two kinds of hoarding behavior, namely high-frequency and low-frequency hoarding [38]. In this study, Brandt’s voles showed an obvious individual division of labor as there were high- and low-frequency hoarders, thus verifying the hypothesis of this study. High-frequency hoarders spent more time on food hoarding and sorting, but the hoarded food was made available for all individuals living in the same burrow system. In the face of predation risks in nature, members need to share vigilance responsibilities and other work [39]. However, in this study, low-frequency hoarders were not found to have greater behavioral ability in other divisions of labor, but this may have been due to the limited range of the video cameras. Although all individuals in the study area were stained and marked in this study, it was difficult to fully capture the free activities of all voles in the burrow system.

Many studies have shown that an increase in food can change the foraging behavior of rodents [40]. The factors affecting animal food selection are both complex and accidental. Animals in the wild are influenced by the nutritional quality, availability, and predation risk of food in their food selection [41]. In order to reduce the risk of predation, small rodents are usually more inclined to forage in the area close to the hole. In this study, in the added food treatment, peanuts were placed near the hole, which enabled Brandt’s voles to obtain high-quality food near the hole. The studies on Korean field mice (*Apodemus peninsulae*) and Rock squirrels (*Sciurotamias davidianus*) showed, respectively, that the hoarding behavior changed under the conditions of different competitive interference, and that different strategies were adopted by the animals to obtain more food resources [42]. Similarly, the increased hoarding frequency in the added food and added food + competition treatments suggests that food availability is a critical factor inducing hoarding behavior. The competition for food resources may affect the hoarding behavior of Brandt’s voles, which was also consistent with the hypotheses of this study.

Food hoarding is equally important for both male and female animals, and the sex difference in animal hoarding behavior has its evolutionary adaptive significance [16]. Sex hormones may be one of the causes of sex differences in hoarding behavior [43]. Although females generally hoard more food than males under non-interfering competition conditions in the Eurasian red squirrel *Sciuras vulgaris* [44], the role of sex in food hoarding has rarely been reported under interfering competition. When male and female Woodland voles (*Microtus pinetorum*) are reared together, females store almost no food, while males store a lot of food [45]. However, in a study on Golden hamsters, Forest buckthorn voles (*Heteromys desmarestianus*), Mongolian gerbils, Spiny squirrels (*Liomys salvini*), and Red squirrels (*Tamiasciurus hudsonicus*), it was found that the food-storing behavior was greater among female individuals and weaker among male individuals [12,43]. However, this study found no significant difference in the frequency of food hoarding between male and female Brandt’s voles, which may have been related to the fact that Brandt’s voles enter the food-hoarding period in autumn when their reproductive organs shrink and therefore both males and females are involved in food hoarding. Further, food hoarding occurs at a higher frequency during the autumn than in other seasons. The previous findings in Mongolian gerbils, which revealed cluster hoarding [46] and no sex difference in food hoarding [47], support our current results in the voles.

Food hoarding is often accompanied by certain behavioral characteristics (such as inquiry ability and memory ability). No significant difference in inquiry ability was observed between high- and low-frequency food hoarders, which was inconsistent with our hypothesis. This may be attributed to the different behavior patterns of scatter hoarders and concentrated hoarders. The scatter hoarders establish many scattered hoarding points during the hoarding process [48], and they rely on greater spatial inquiry ability to locate these hoarding points during the long winter [49]. In contrast, Brandt’s voles mainly showed gregarious and concentrated food hoarding. The ecological adaptation of Brandt’s voles to concentrate on food hoarding may be weakly related to their exploration behavior. Zhang et al. performed a comparative study on the exploration behavior of rodents with four different food-hoarding methods and also supported the above conclusions. They found that Kunming mice (*Mus musculus*), which displayed concentrated food hoarding, had weaker exploration behavior than Korean field mice, which display dispersed food hoarding [50].

The high-frequency hoarders of Brandt’s voles had better spatial memory than the low-frequency hoarders. The ability to learn and remember enables animals to adapt to a changing environment [51]. Animals with greater spatial memory can quickly locate information related to food and predators, which is conducive to their survival and reproduction. For example, an instinctive defense depends on the rapid learning of spatial environment information [52]. Brandt’s voles live in a vast steppe area with a complex living environment. They need to go out to forage in the steppe environment, as well as to carry out social interactions and resist natural enemy predation. Therefore, Brandt’s voles with high food reserves can adapt to higher environmental selection pressure and thereby evolve to improve their ability to learn and remember.

## 5. Conclusions

In summary, Brandt’s voles showed an obvious food-hoarding division of labor, with a distinction between individuals who displayed a high frequency of food hoarding and a low frequency. At the same time, high- and low-food-hoarding individuals did not show a division of labor in other behaviors. Food-hoarding behavior is not related to body weight, sex, individual inquiry behavior, or the division of food hoarding, but relies on the ability to learn and remember. This difference in learning and memory ability may be a key factor in the division of labor behavior of Brandt’s voles.

## Figures and Tables

**Figure 1 animals-14-02719-f001:**
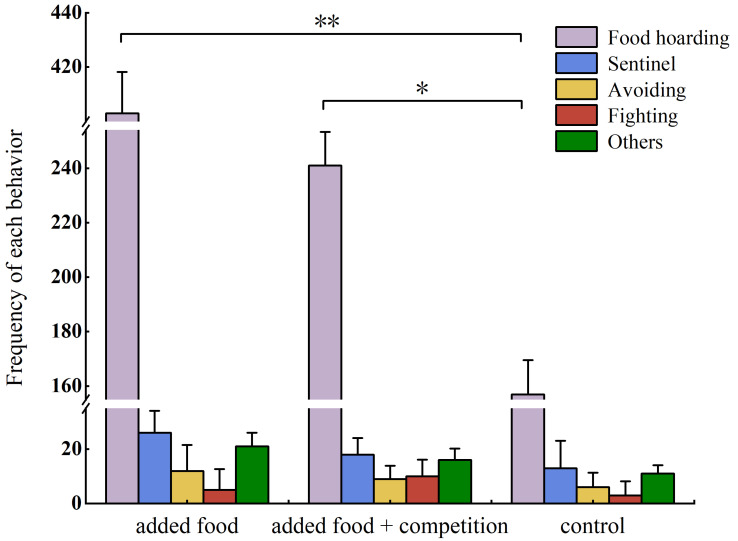
Behavioral allocation of Brandt’s voles. Note: The same letter indicates the absence of significant differences. * *p* < 0.05, ** *p* < 0.01. Data are presented as mean ± SE.

**Figure 2 animals-14-02719-f002:**
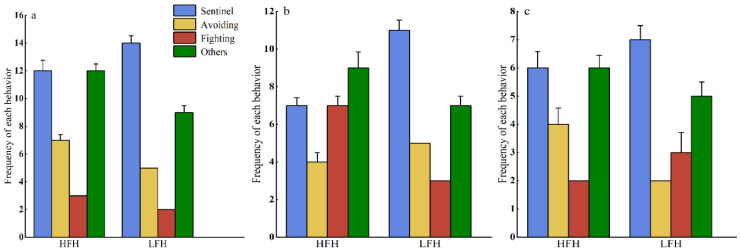
Comparison of the behavior of Brandt’s voles in the high-food-hoarding group (HFH) and the low-food-hoarding group (LFH). There was no significant difference between the high- and low-food-hoarding groups in the added food treatment (**a**), the added food + competition treatment, (**b**) or the control treatment (**c**). Data are presented as mean ± SE.

**Figure 3 animals-14-02719-f003:**
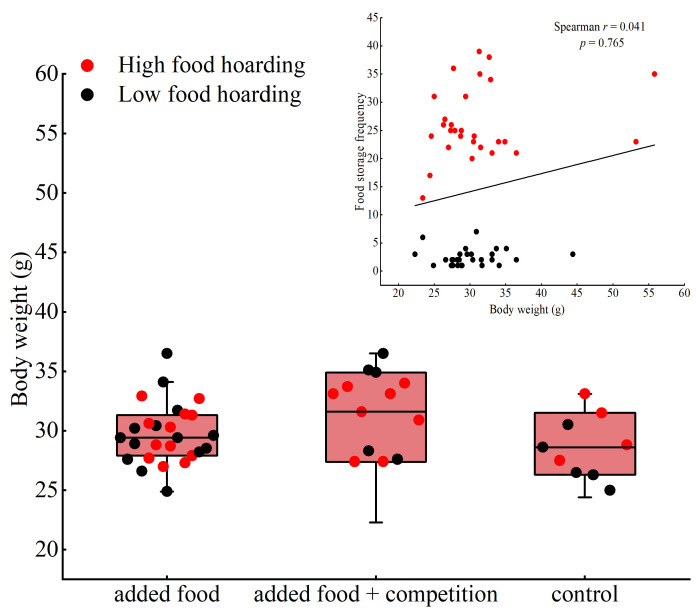
The relationship between food-hoarding frequency and body weight in different treatments.

**Figure 4 animals-14-02719-f004:**
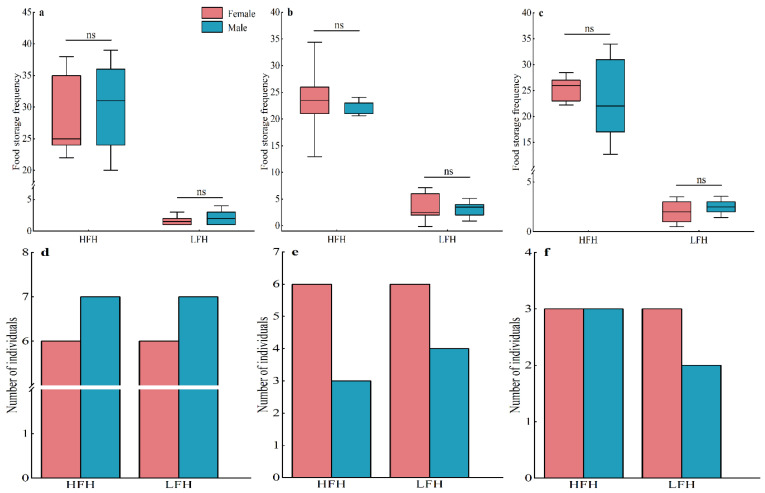
The individual differences in the frequency of food hoarding and between males and females in high-food-hoarding (HFH) and low-food-hoarding (LFH) individuals in each treatment; The frequency of food hoarding between males and females in the high- and low-food-hoarding groups in the added food treatment (**a**), the added food + competition treatment (**b**), and the control treatment (**c**); the number of male and female individuals in the high- and low-food-hoarding groups in the added food treatment (**d**), the added food + competition treatment (**e**), and the control treatment (**f**). Data are presented as mean ± SE; ns indicates that the difference is not significant.

**Figure 5 animals-14-02719-f005:**
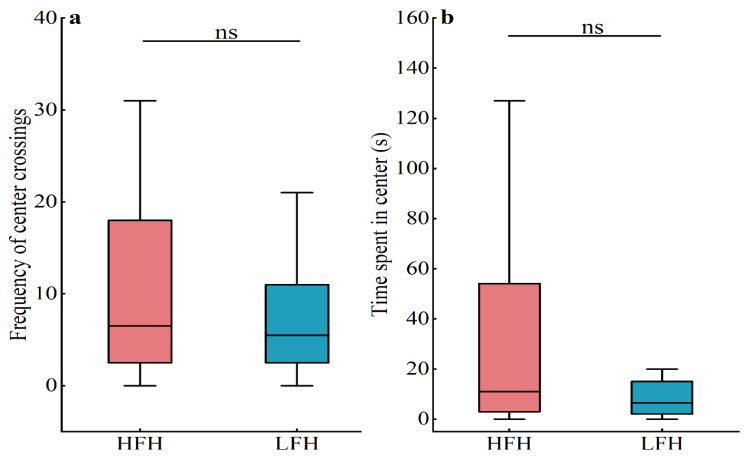
The exploratory behavior between high-food-hoarding (HFH) and low-food-hoarding (LFH) individuals: (**a**) the frequency of center crossings; (**b**) the time spent in the center. Data are presented as mean ± SE; ns indicates that the difference is not significant.

**Figure 6 animals-14-02719-f006:**
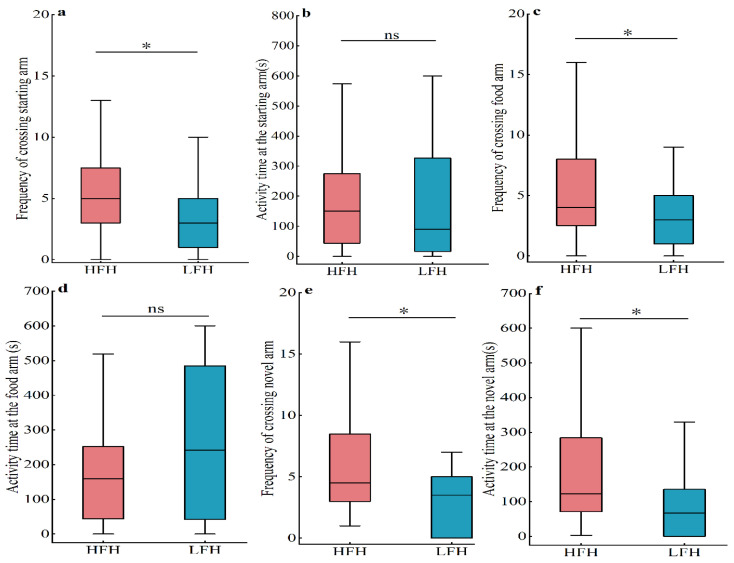
Comparison of spatial memory between high-food-hoarding (HFH) and low-food-hoarding (LFH) individuals. The frequency of crossing the starting arm (**a**), the time spent in the starting arm (**b**), the frequency of crossing the food arm (**c**), the time spent in the food arm (**d**), the frequency of crossing the novel arm (**e**), and the time spent in the novel arm (**f**). Data are presented as mean ± SE. * *p* < 0.05; ns indicates that the difference is not significant.

**Table 1 animals-14-02719-t001:** Brandt’s voles behavior spectrum.

Behavior Type	Description
Foraging	During the movement, performing a series of actions to search for food such as crawling, smelling the ground and vegetation, and digging at grass roots.
Feeding–Food hoarding	While situated in the posture of standing, sitting, squatting, or lying down, the two front claws grab food to assist the mouth in the biting and chewing of food. Bringing the food back to the warehouse in the nest so that it can be eaten when food is scarce.
Chirping	After discovering danger, examining the dangerous object while sitting, standing, or squatting, and issuing a series of screams to warn similar individuals.
Sentinel avoiding	Interrupting the ongoing behavior (such as running, feeding, foraging, etc.) to squat or sit still while rapidly turning the head to observe the surrounding environment to determine whether there is danger around, generally does not take longer than 3 s. Interrupting ongoing behavior (such as running, feeding, foraging, etc.) when finding danger or hearing a call and quickly running back into the hole, sometimes with a scream.
Fighting	A scuffle between two individuals.

**Table 2 animals-14-02719-t002:** Number of marked individuals after live-capture and captured on video.

	Added Food		Added Food + Competition		Control	
	Female	Male	Female	Male	Female	Male
Marked after capture	23	14	22	9	13	9
Captured on video	12	14	12	7	6	5

**Table 3 animals-14-02719-t003:** Differences in the hoarding frequency of Brandt’s voles in each treatment.

Sample Area	Added Food	Added Food + Competition	Control
Food-hoarding frequency	15.50 ± 14.60	12.74 ± 11.00	14.27 ± 12.06
	*H* = 5.350	*H* = 5.536	*H* = 0.468
Significance level	*p* = 0.069	*p* = 0.063	*p* = 0.791

## Data Availability

The raw data supporting the conclusions of this article will be made available by the authors, without undue reservation.

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
