# Peer review of "The Individual Division of Food Hoarding in Autumn Brandt’s Voles (Lasiopodomys brandtii)"

_animals, 2024, doi:10.3390/ani14182719_

Round 1

Reviewer 1 Report

Comments and Suggestions for Authors

Interesting study, well written paper. The only thing I'd change is replace the term gender (typically used to refer to gender identity in people) with the term sex (biological sex in animals). 

Author Response

Comment 1: Interesting study, well written paper. The only thing I'd change is replace the term gender (typically used to refer to gender identity in people) with the term sex (biological sex in animals). 

Response 1: Thank you very much for your recognition of the article and the valuable suggestions. We replace “gender” with “sex” in lines 49,84,107,146,231,282,362,364,366,379 and 408 of the manuscript.

Reviewer 2 Report

Comments and Suggestions for Authors

This manuscript is well organized and written. There is good use of the scientific literature. The figures and tables are appropriate and useful. The paper does a good job of showing various attributes, activities and behaviors of various vole individuals under different settings. I only had one possible suggestion for improving the manuscript.  Perhaps a diagram of the Y maze and where the vole starts in the maze? Does the vole start at the middle of the maze, or at the start of the start arm. The way we used Y mazes, the rodent was placed at the start of the start arm and then could walk along until it came to the Y at which point it could go into the right arm or the left arm (each of which contained a different lure/stimulus). 

Author Response

This manuscript is well organized and written. There is good use of the scientific literature. The figures and tables are appropriate and useful. The paper does a good job of showing various attributes, activities and behaviors of various vole individuals under different settings. I only had one possible suggestion for improving the manuscript.  Perhaps a diagram of the Y maze and where the vole starts in the maze? Does the vole start at the middle of the maze, or at the start of the start arm. The way we used Y mazes, the rodent was placed at the start of the start arm and then could walk along until it came to the Y at which point it could go into the right arm or the left arm (each of which contained a different lure/stimulus). 

Response 1: Thank you very much for your recognition and valuable suggestions on this article. Sorry, this part is not clearly stated in the manuscript. At the beginning of the experiment, Brandt voles were placed at the starting point of the starting arm and allowed to move freely. We provided a detailed description in lines 215-216 of the manuscript.

Reviewer 3 Report

Comments and Suggestions for Authors

See attached file.

Comments on the Quality of English Language

I believe the manuscript would greatly benefit from a thorough editing regarding grammar and syntax to improve the overall readability.  
